


# Rapid adaptive Optimization Model for Atmospheric Chemistry (ROMAC) v1.0

Jiangyong Li[1,2], Chunlin Zhang[1,2], Wenlong Zhao[1], Shijie Han[1,2], Yu Wang[1,3], Hao Wang[1,2]*, Boguang Wang[1,2]*

*Correspondence to*: Boguang Wang (tbongue@jnu.edu.cn) Hao Wang (wanghao@jnu.edu.cn)

[1] Australia-China Centre for Air Quality Science and Management (Guangdong), Institute for Environmental and Climate Research, Jinan University, Guangzhou, 511443, China

[2] Guangdong Provincial Observation and Research Station for Atmospheric Environment and Carbon Neutrality in Nanling Forests, Guangzhou, 511443, China

[3] Air Quality Studies, Department of Civil and Environmental Engineering, The Hong Kong Polytechnic University, Hong Kong, China

**Abstract.** Rapid adaptive Optimization Model for Atmospheric Chemistry (ROMAC) is a flexible and computationally efficient photochemical box model. The unique adaptive dynamic optimization module in ROMAC enables it to dynamically and rapidly estimate the impact of chemical and physical processes on pollutant concentration. ROMAC overcomes the shortcomings of over-simplified physical modules in traditional box models, and its ability to quantify the effects of chemical and physical processes on pollutant concentrations has been confirmed by the chamber and field observation cases. Since a variable step and variable order numerical solver without Jacobian matrix processing was developed, the computational efficiency of ROMAC is significantly improved. Compared with other box models, the computational efficiency of ROMAC is improved by 96%.

**Keywords** Atmospheric chemistry, Photochemical box model, Numerical simulation, High computational efficiency, Stiff ODEs

## 1 Introduction

Numerical models are effective tools of atmospheric chemistry studies. The 0-dimensional box model has been widely used in previous studies to study the relationship between secondary pollutants and precursors (Decker et al., 2021; Decker et al., 2019; Ling et al., 2017; Wang et al., 2017; He et al., 2019). Box model can be used as a ground Lagrangian trajectories model to study the influence of regional transport of precursors on the formation of secondary pollutants (Cheng et al., 2010; Wang et al., 2019). In addition, the box model is also a powerful tool in environmental chamber studies (Chen et al., 2015; Novelli et al., 2018).





Since the processes of vertical and horizontal transmission are ignored, the simulation speed of the 0-D box model is higher than that of the 3D air quality model. This allows box models to use more comprehensive chemical mechanisms, and focusing on the analysis of chemical processes. However, with the development of atmospheric chemistry mechanism, the number of chemical reactions involved gradually increases, and the simulation of 0-D box model is still a time-consuming process when

using chemical mechanisms with a large number of reactions. Chemical transformations can be described by a series of ordinary differential equations (ODEs), and solving the numerical solution of the ODEs is one of the time-consuming tasks of the box model. For example, the Master Chemical Mechanism (MCM v3.3.1) contains about 5900 species (Jenkin et al., 2015), and the size of the Jacobian matrix is close to 5900×5900, which requires a large number of matrix calculations in the process of solving with the implicit solver. Therefore, it is necessary to develop a computationally efficient model for chemical

mechanisms.

Several box models have been developed and applied in previous studies, such as AtChem (Sommariva et al., 2020), Chemistry As A Box Model Application (CAABA) (Sander et al., 2011), Framework for 0-D Atmospheric Modeling (F0AM) (Wolfe et al., 2016) , PyCHAM (O'meara et al., 2021), JlBox (Huang and Topping, 2021) and PBM-MCM (Wang et al., 2018). Most of these models rely on third-party tools for differential equation solving. For example, AtChem uses the CVODE library

to integrate the ODEs of chemical mechanism. F0AM uses ode15s in MATLAB, a variable order and variable time step solver based on Gear's method. FACSIMILE was used to integrate the ODEs in the PBM-MCM model. Several multistep or multistage approaches are commonly used by these chemical solvers, such as ROSENBROCK, BDF, LSODE, GEAR, SMVGEAR, etc. (Verwer et al., 1996; Aro, 1996b; Sandu et al., 1997a; Sandu et al., 1997b). Although these solving tools have good accuracy and stability, the solving process requires a lot of computing resources, which significantly reduces the

computational efficiency.

Previous studies have developed several approaches to improve the efficiency of simulation. One way to improve the computational efficiency is to simplify the chemical mechanism, such as SAPRC07 (Carter, 2012) and CB6 (Yarwood, 2010), which are commonly used in 3D air quality models. The MCM mechanism also has a simplified version (http://cri.york.ac.uk/), which can improve the computational efficiency. However, the simplified mechanism will lead to bias in the simulation results

of radicals (*e.g.,* OH, $HO_2$, $RO_2$) and secondary pollutant concentration(Ying and Li, 2011; Jimenez, 2003). Another approach is to improve the computational efficiency of differential equation solver, such as using GPU acceleration(Alvanos and Christoudias, 2017) or using quasi-Newton method (Esentürk et al., 2018). These methods can effectively shorten the running time of the program, but still need to consume a lot of memory, CPU or GPU resources when processing the Jacobian matrix. There are also solution methods that do not need to store and update the Jacobian matrix, such Quasi-steady State

Approximation (QSSA), multistep explicit and semi-implicit methods (Mott et al., 2000; R. and Boris, 1977). But these





methods usually do not conserve mass (Cariolle et al., 2017). There are also fully implicit methods that do not need to deal

with the Jacobian matrix, such as Euler Backward Iterative (EBI) (Hertel et al., 1993). EBI method is widely used in 3D

chemical transport model (*e.g.,* Community Multiscale Air Quality model, Nested Air Quality Prediction Modeling System)

because it is computationally efficient. But EBI solver has a large truncation error because it is only first-order accurate.

Another stiff ODEs preconditioner method based on Newton linearization, also simplifies the matrix operations during the

solution (Aro, 1996b). However, these algorithms may fail to converge when the Jacobian matrix is significantly off-diagonally

dominant (Aro, 1996a). Hence, with the increasing of complexity and scale of chemical mechanism systems, it is still a

challenge to make these solving algorithms converge stably.

Rapid adaptive Optimization Model for Atmospheric Chemistry (ROMAC) is a computationally efficient photochemical

box model. A variable-step and variable-order solver without Jacobian matrix processing is developed for the ROMAC model.

Since the ROMAC model is computationally efficient, accurate and stable, users can dynamically optimize the influence of

physical processes on pollutant concentration, and overcome the shortcomings of the lack of physical processes in the

traditional box models.

## 2 Description of the ROMAC model

ROMAC is a 0-D model focused on the simulation of atmospheric chemical kinetics problem. It was developed to provide

users with a flexible and efficient computational tool. The core modules of ROMAC were developed in Fortran, and the data

pre-processing and post-processing modules were developed in python, which can keep the model running efficiently and

provide users with flexible processing tools. In ROMAC, the changes in concentration of a species can mathematically be

represented as Eq. (1).

$$\frac{dc}{dt} = [\frac{dc}{dt}]_{chem} + [\frac{dc}{dt}]_{emis} + [\frac{dc}{dt}]_{dry} + [\frac{dc}{dt}]_{dilu} + [\frac{dc}{dt}]_{others} \tag{1}$$

Where $[\frac{dc}{dt}]_{chem}$ represent the changes due to chemical reactions; $[\frac{dc}{dt}]_{emis}$ represents the emission rate for the species; $[\frac{dc}{dt}]_{dry}$

and $[\frac{dc}{dt}]_{dilu}$ represent the dry deposition and dilution, respectively. For dry deposition, ROMAC uses the maximum dry

deposition velocity (cm s[-1]) calculated by Zhang et al (2003)   to estimate the dry deposition process of the species, and user

can also customize this value. The dry deposition process is added to the model in the form of first-order kinetics, and the

kinetic constant is calculated by the dry deposition velocity and the preset boundary layer height (cm). For dilution, similar to

other models (Wolfe et al., 2016; Sommariva et al., 2020), ROMAC uses first-order kinetics to calculate the dilution process,

and users can customize the constants of the dilution process.





In addition, ROMAC model sets a user-defined term of rate ($[\frac{dc}{dt}]_{others}$), and the user can add additional change rates if needed, such as the gas-wall partitioning in the chamber studies and the external transport in field observations. Users can also define $[\frac{dc}{dt}]_{others}$ as physical processes (*e.g.,* vertical and horizontal transport), and estimate the contribution of the physical

process to the pollutant concentration by the dynamical optimization algorithm. It helps to overcome the shortcomings of the over-simplified physical modules in the traditional box models.

**2.1 High efficiency solver for atmospheric chemical kinetic equations**

Unlike most existing models, ROMAC does not rely on third-party libraries for numerical solving. ROMAC has its own computationally efficient numerical solver. The solver in ROMAC is optimized according to the characteristics of the

atmospheric chemical mechanism, and it will be a universal chemical solver.

Chemical mechanism is the core of atmospheric chemical box model. Generally, chemical reaction equations can be described in Eq. (2).

$$\alpha_1 r_1 + \alpha_2 r_2 + \dots + \alpha_n r_n \rightarrow \beta_1 p_1 + \beta_2 p_2 + \dots + \beta_m p_m \tag{2}$$

Where $\alpha$ and $\beta$ represents stoichiometric number, r and p represent reactant and product, respectively. Hence, derivative of species concentration with respect to time can be described as an ODEs system shown in Eq. (3). For specie *i*, $f_i$ can be

calculated by Eq. (4). In Eq. (4), $P_{i,t}$ and $L_{i,t}$ denote the chemical generation rate and the loss rate of species *i* at time *t*, respectively. It is worth to note that the loss rate is related to the concentration of species *i*. Therefore, to facilitate the subsequent formula derivation, $L_{i,t}$ can be described as a multivariate higher-degree equations for the concentration of species *i* shown in Eq.(5). Where $R_{tot}$ represents the number of the reactions related to the loss rate of species *i*; $\alpha$ is the stoichiometric number, and $l_{i,t,R}$ is the part of the chemical reaction rate that is not directly related to the concentration of species *i*. The

computation of the $f$ ($C_t$, *t*) follows the approach in the Fortran code provided by MCM official website (http://mcm.york.ac.uk/extract.htt).

$$[\frac{dC_t}{dt}]_{chem} = f(C_t, t) \tag{3}$$

$$f_i(C_{i,t}, t) = P_{i,t} - L_{i,t} \tag{4}$$

$$L_{i,t} = \sum_{R=1}^{R_{tot}} l_{i,t,R} C_{i,t}^{\alpha_R} \tag{5}$$

The lifetime of different species in atmospheric chemical mechanism varies greatly. For example, OH has an atmospheric lifetime of only seconds but $O_3$ has a lifetime of several days. Therefore, the ODEs system of atmospheric chemical kinetics simulation is extremely stiff, and explicit methods (*e.g.,* explicit Euler method, explicit Runge-Kutta method) are difficult to

solve these problems.



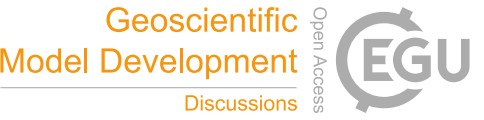

In ROMAC, an implicit Euler method was used to solved the ODEs, the iteration formula is given in Eq. (6). Due to its superior numerical stability, this method has been widely used in other atmospheric chemistry models (Esentürk et al., 2018). However, due to the implicit Euler method only has first-order accuracy, it may introduce large truncation errors in the process of integration. Hence, the trapezoidal method iteration formula shown in Eq. (7) is used for integration in a specific situation.

Both implicit Euler method and trapezoidal method have the term of $f(C_{t+1}, t+1)$ which is unknown at time $t$ and needs to be solved. Newton–Raphson (NR) scheme is a popular method for solving such implicit equations. Newton–Raphson scheme will be described by Eq. (6) to Eq. (12). Equations (6) and (7) can be expressed in the form of Eq. (8) and Eq. (9), respectively.

$$C_{t+1} = C_t + f(C_{t+1}, t+1)\Delta t \tag{6}$$

$$C_{t+1} = C_t + \frac{f(C_t, t) + f(C_{t+1}, t+1)}{2}\Delta t \tag{7}$$

$$g_1(C_{t+1}) = C_{t+1} - C_t - f(C_{t+1}, t+1)\Delta t = 0 \tag{8}$$

$$g_2(C_{t+1}) = C_{t+1} - C_t - \frac{f(C_t, t) + f(C_{t+1}, t+1)}{2}\Delta t = 0 \tag{9}$$

So, the iteration formula can be expressed in the form of Eq. (10). Where $\nabla g^{-1}(C_{t+1})$ the inverse matrix of the Jacobian matrix of $g(C_{t+1})$. The Jacobian matrix for the implicit Euler method is given in Eq. (11), and the Jacobian matrix for the

trapezoidal method is given in Eq. (12). It should be noted that the size of Jacobian matrix and its inverse matrix will increase with the number of species in chemical mechanism increasing. In particular, dealing with explicit chemical mechanisms (*e.g.,* MCM) would consume a lot of computer resources to store the Jacobian matrix and its inverse matrix. In addition, the inverse of a large-scale Jacobian matrix is quite time-consuming.

$$C_{t+1}^{k+1} = C_{t+1}^k - \nabla g^{-1}(C_{t+1})g(C_{t+1}) \tag{10}$$

$$\nabla g_1(C_{t+1}) = \begin{bmatrix} 1 - \frac{\partial f_1(C_{1,t+1})}{\partial C_{1,t+1}}\Delta t & \cdots & -\frac{\partial f_1(C_{1,t+1})}{\partial C_{n,t+1}}\Delta t \\ \vdots & \ddots & \vdots \\ -\frac{\partial f_n(C_{n,t+1})}{\partial C_{1,t+1}}\Delta t & \cdots & 1 - \frac{\partial f_n(C_{n,t+1})}{\partial C_{n,t+1}}\Delta t \end{bmatrix} \tag{11}$$

$$\nabla g_2(C_{t+1}) = \begin{bmatrix} 1 - \frac{\partial f_1(C_{1,t+1})}{\partial C_{1,t+1}}\Delta t - \frac{\partial L_{1,t+1}}{\partial C_{1,t+1}}\frac{\Delta t}{2} & \cdots & -\frac{\partial f_1(C_{1,t+1})}{\partial C_{n,t+1}}\Delta t + \frac{\partial P_{1,t+1}}{\partial C_{n,t+1}}\frac{\Delta t}{2} - \frac{\partial L_{1,t+1}}{\partial C_{n,t+1}}\frac{\Delta t}{2} \\ \vdots & \ddots & \vdots \\ -\frac{\partial f_n(C_{n,t+1})}{\partial C_{1,t+1}}\Delta t + \frac{\partial P_{n,t+1}}{\partial C_{1,t+1}}\frac{\Delta t}{2} - \frac{\partial L_{n,t+1}}{\partial C_{1,t+1}}\frac{\Delta t}{2} & \cdots & 1 - \frac{\partial f_n(C_{n,t+1})}{\partial C_{n,t+1}}\Delta t - \frac{\partial L_{n,t+1}}{\partial C_{n,t+1}}\frac{\Delta t}{2} \end{bmatrix} \tag{12}$$

A Simplified-Newton (SN) method can effectively reduce the computational complexity of the iterative process of NR

method. Traditional SN method substitute the inverse Jacobian matrix obtained in the first iteration for the inverse matrix in the subsequent iterations. Although the traditional SN method can reduce the amount of computation, it still needs to calculate and store the inverse of the Jacobian matrix at each time step. To further improve the computational efficiency, ROMAC uses a Diagonal-Simplified-Newton (DSN) method to solve the implicit equations.





When the $\Delta t$ in Eq. (11) and Eq. (12) is small enough, the Jacobian matrix of $g(C_{t+1})$ will be a diagonally dominant matrix

or a quasi-diagonally dominant matrix. Under these conditions, the inverse matrix of Jacobian can be approximated by Eq.

(13). According to the equations associated with the implicit Euler method in Eq. (1) to Eq. (13), the iteration formula for

specie $i$ is shown in Eq. (14). Where $k$ represents the number of iterative solutions. Previous study has also shown that such

approximations are reliable (Aro, 1996a). Similarly, the approximate inverse of the Jacobian matrix for the trapezoidal method

and the iterative formulas for the solution can be derived as shown in Eq. (15) and Eq. (16), respectively.

$$\nabla g_1^{-1}(C_{t+1}) \approx \begin{bmatrix} \dfrac{1}{1-\dfrac{\partial f_1(C_{1,t+1})}{\partial C_{1,t+1}}\Delta t} & \cdots & 0 \\ \vdots & \ddots & \vdots \\ 0 & \cdots & \dfrac{1}{1-\dfrac{\partial f_n(C_{n,t+1})}{\partial C_{n,t+1}}\Delta t} \end{bmatrix} \tag{13}$$

$$C_{i,t+1}^{k+1} = \frac{\sum_{R=1}^{R_{tot}}(\alpha_R-1)l_{t+1,R}C_{i,t+1}^{k}{}^{\alpha_R}\Delta t + C_{i,t} + P_{t+1}\Delta t}{1+\sum_{R=1}^{R_{tot}}\alpha_R l_{t+1,R}C_{i,t+1}^{k}{}^{\alpha_R-1}\Delta t} \tag{14}$$

$$\nabla g_2^{-1}(C_{t+1}) \approx \begin{bmatrix} \dfrac{1}{1-\dfrac{\partial f_1(C_{1,t+1})}{\partial C_{1,t+1}}\Delta t - \dfrac{\partial L_{t+1}}{\partial C_{1,t+1}}\dfrac{\Delta t}{2}} & \cdots & 0 \\ \vdots & \ddots & \vdots \\ 0 & \cdots & \dfrac{1}{1-\dfrac{\partial f_n(C_{n,t+1})}{\partial C_{n,t+1}}\Delta t - \dfrac{\partial L_{t+1}}{\partial C_{n,t+1}}\dfrac{\Delta t}{2}} \end{bmatrix} \tag{15}$$

$$C_{i,t+1}^{k+1} = \frac{\sum_{R=1}^{R_{tot}}(\alpha_R-1)l_{t+1,R}C_{i,t+1}^{k}{}^{\alpha_R}\Delta t + 2C_{i,t} + P_{i,t}\Delta t - L_{i,t}\Delta t + P_{i,t+1}\Delta t}{2+\sum_{R=1}^{R_{tot}}\alpha l_{t+1,R}C_{i,t+1}^{k}{}^{\alpha_R-1}\Delta t} \tag{16}$$

It's worth noting that if all of the stoichiometric number ($\alpha_R$) is equal to 1, Eq.(14) is the same as the iteration formula of EBI

solver (Hertel et al., 1993) used in CMAQ model. In this study, Eq. (14) provides a generalized form of the EBI iteration

formula. Hertel's (1993) study shows that EBI solver has the advantages of high computational efficiency and high accuracy.

However, the convergence condition of this method has not been discussed, such as how to choose the optimal integration

time step size to make the solution process stable and convergent. If the time step size was too short, the computational

efficiency will decrease. However, if the time step size is too large, the Jacobian matrix will not be diagonally dominant, it

will lead algorithm hard to converge or even not converge. This problem also exists in the EBI algorithm. Especially for such

a complex chemical mechanism as MCM, directly using the EBI scheme will have a large risk of causing the algorithm not to

converge. In ROMAC, a variable time step and variable order scheme was developed to balance the computational efficiency

and accuracy. The variable time step scheme can also maintain the Jacobian matrix as a quasi-diagonally dominant matrix and

reduce the risk of convergence failure. Hence, the numerical solver in ROMAC model will overcome the shortcoming of EBI

solver.

Actually, it is difficult to use a fixed time step to ensure that the Jacobian matrix is always quasi-diagonally dominant. In

order to find the optimal time step, a variable time step size scheme is used in our model. First, $\Delta t_0$ is defined as an extremely



small positive value to ensure that this value is not less than the rounding error of the computer. According to IEEE Std 754-2008 (Committee, 2008), $\Delta t_0$ is defined as $2.22 \times 10^{-16}$ seconds in ROMAC. Secondly, $\Delta t_1$ is defined as atmospheric lifetime of the species with the shortest lifetime in the chemical mechanism, as shown in Eq. (17). Third, a strict diagonal dominance matrix requires that the diagonal elements are greater than the sum of the rest of the elements in the same row, as shown in Eq. (18). Hence, $\Delta t_{2,i}$ is calculated by Eq.(19) to ensure that Eq.(18) holds, and $\Delta t_2$ is the minimum in the set of $\Delta t_{2,i}$ shown in Eq.(19). Where $i$ represents the rows of the Jacobian matrix. Finally, the initial integration time step size is determined by Eq.

155 (21).

$$\Delta t_1 = [\frac{1}{L_t}]_{min} \tag{17}$$

$$|\nabla g(C_{t+1})_{i,i}| > \sum_{j=1}^{n} |\nabla g(C_{t+1})_{i,j}| \tag{18}$$

$$\Delta t_{2,i} = \frac{0.9}{(\sum_{j=1}^{n} |\frac{\partial f_i(C_{1,t+1})}{\partial C_{j,t+1}}|)} \tag{19}$$

$$\Delta t_2 = [\Delta t_{2,i}]_{min} \tag{20}$$

$$\Delta t_{init} = \begin{cases} \boldsymbol{\Delta t_0} & (\qquad \Delta t_0 \geq \Delta t_1 \ and \ \Delta t_0 \geq \Delta t_2 \qquad) \\ \boldsymbol{\Delta t_1} & (\Delta t_0 < \Delta t_1 \ and \ \Delta t_0 < \Delta t_2 \ and \ \Delta t_2 \geq \Delta t_1) \\ \boldsymbol{\Delta t_2} & (\Delta t_0 < \Delta t_1 \ and \ \Delta t_0 < \Delta t_2 \ and \ \Delta t_2 < \Delta t_1) \end{cases} \tag{21}$$

In order to improve the computational efficiency, the integration time step size should grow while ensuring the accuracy of the solution. When the time step size grows, the local truncation error (*LTE*) should be controlled. In each step ($\Delta t$), ROMAC model uses both single-step and double-step methods for integration, and the calculated results are recorded as $C_{\Delta t}$ and $C_{\frac{\Delta t}{2}}$, respectively. *LTE* is estimated by the difference between $C_{\Delta t}$ and $C_{\frac{\Delta t}{2}}$ ( $LTE = \left| C_{\frac{\Delta t}{2}} - C_{\Delta t} \right|$ ), and the relative error is

estimated by Eq. (22). This method has been successfully used in previous study (Aro, 1996b).

$$RERR = [\frac{\left| C_{\frac{\Delta t}{2}} - C_{\Delta t} \right|}{1 + C_{\Delta t}}]_{min} \tag{22}$$

The model needs to adjust the integration time step according to the tolerance preset by user. This requires inferencing a maximum integration time step based on the preset tolerance. According to the Lagrange remainder of Taylor formula, the *RERR* of the integration result can also be expressed as Eq. (23). Where $s$ is the order of integration accuracy, equal to 1 for the implicit Euler method and equal to 2 for the trapezoidal method. Similarly, the user-specified maximum integral relative

error can be expressed as Eq. (24). Where $\Delta t_{max}$ is an estimate of the maximum step size allowed when the preset *rtol* condition is satisfied. In ROMAC, the values of $\xi_1$ and $\xi_2$ in Eq. (23) and Eq. (24) are assumed to be approximate ($\xi_1 \approx \xi_2$). According to Eq. (23) and Eq. (24), the maximum integration time step can be estimated by Eq. (25). Finally, the integration time step is



updated according to Eq. (26) and Eq. (27) to make sure that the time step not larger than the maximum time step. In order to

avoid a too accurate result make the integration step size grow too large, when $\Delta t_{opt}$ is greater than $\Delta t$ by a factor of 10, the

time step is only increased by a factor of 10. In general, as the integration step size increases, the number of iterations (N)

required by the solver in this study will also increase. Too many iterations will make the computation time-consuming, so the

integration time step is not increased when the solver iteration time exceeds 50 (N ≥ 50).

$$RERR = \frac{R_n(\Delta t)}{1 + C_{\Delta t}} = \frac{f^{(s+1)}(\xi_1)\Delta t^{s+1}}{(1 + C_{\Delta t}) \times (s+1)!} \tag{23}$$

$$rtol = \frac{R_n(\Delta t_{max})}{1 + C_{\Delta t}} = \frac{f^{(s+1)}(\xi_2){\Delta t_{max}}^{s+1}}{(1 + C_{\Delta t}) \times (s+1)!} \tag{24}$$

$$\Delta t_{max} = \left(\frac{rtol}{RERR}\right)^{\frac{1}{s+1}}\Delta t_t \tag{25}$$

$$\Delta t_{opt} = 0.9\Delta t_{max} \tag{26}$$

$$\Delta t_{t+1} = \begin{cases} \Delta t_{opt} & (\Delta t_{opt} < 10\Delta t_t \ and \ N < 50) \\ 10\Delta t_t & (\Delta t_{opt} \geq 10\Delta t_t \ and \ N < 50) \\ \Delta t_t & (N \geq 50) \end{cases} \tag{27}$$

ROMAC model has a strict control on the truncation error of integration according to the relative tolerance (*rtol*) and the

absolute tolerance (*atol*) specified by the user. If *RERR* < *rtol* or *LTE* < *atol*, proceed to the next integration time, otherwise

the integration time step is halved and re-integrated until the tolerance requirement is satisfied.

Another important question is whether to choose the implicit Euler method or the trapezoidal method for the integration

process. Both implicit Euler method and trapezoidal method are stable for stiff ODEs. However, the solution method used in

this study requires the Jacobian matrix to be diagonally dominant or quasi-diagonally dominant. Since the initial time step in

Eq. (18) and Eq. (19) are derived from the implicit Euler method, the implicit Euler method is used for integral starting. If the

algorithm converges quickly (N < 50), then the trapezoidal method is used on the next integration time step. When N is greater

than 50, the algorithm is switched to implicit Euler method on the next integration time step to improve the computational

efficiency.

The solver for ROMAC uses a variable-step and variable-order approach (VSVOR) to solve stiff ODEs problems. Most of

the time, the accuracy of VSVOR is second order. The VSVOR solver has comparable computational efficiency with the EBI

solver, and the solution accuracy and stability are better.

**2.2 Adaptive dynamic optimization module and variables constraints**

ROMAC can be run under user-specified variable constraints, including but not limited to concentrations of chemical species,

photolysis rate, temperature, humidity, pressure and other meteorological conditions. For concentrations of chemical species,

ROMAC provides the user with three different constraint schemes.





**Scheme 1:**

Different from previous models, ROMAC provides a novel constraint scheme to use the observed data to constrain model run.

Scheme 1 does not directly input the species concentration, but control the $[\frac{dc}{dt}]_{others}$ term with adaptive dynamic optimization

algorithm. The default value for $[\frac{dc}{dt}]_{others}$ is 0, after integration, $\Delta[\frac{dc}{dt}]_{others}$ can be estimated by the gap between the

observed and simulated values, as detailed in Eq. (28).

$$\Delta[\frac{dc}{dt}]_{others} = \begin{cases} \dfrac{C_{obs,n+1} - C_{model,n+1}}{t_{n+1} - t_n} & (|C_{obs,n+1} - C_{model,n+1}| \le 0.1 \times |C_{model,n+1}|) \\ \dfrac{0.1 \times C_{model,n+1}}{t_{n+1} - t_n} \times (-1)^u & (|C_{obs,n+1} - C_{model,n+1}| > 0.1 \times |C_{model,n+1}|) \end{cases} \tag{28}$$

Where $C_{obs}$ represents the observations and $C_{model}$ represents the simulations. In Eq. (28), $u = 1$ when $C_{obs,n+1}$ is less than

$C_{model,n+1}$ and $u = 2$ when $C_{obs,n+1}$ is greater than $C_{model,n+1}$. In complex systems, changing $[\frac{dc}{dt}]_{others}$ may also affect

other chemical process, so the relationship between $[\frac{dc}{dt}]_{others}$ and simulation results may be nonlinear. Therefore, it is difficult

to calculate $[\frac{dc}{dt}]_{others}$ in a single iteration. It is necessary to estimate by loop iteration until the difference between observation

and simulation reaches a preset tolerance. In this study, the difference between observation and simulation is characterized by

Root Mean Square Error (*RMSE*) shown in Eq. (29).

$$RMSE = \sqrt{(C_{obs,n+1} - C_{model,n+1})^2} \tag{29}$$

The cyclic dynamically optimization process of $[\frac{dc}{dt}]_{others}$ is shown in Figure 1. The iterative updating formula of $[\frac{dc}{dt}]_{others}$

based on Newton–Raphson method is given in Eq. (30).

$$[\frac{dc}{dt}]_{others_{m+1}} = \begin{cases} [\frac{dc}{dt}]_{others_m} + \Delta[\frac{dc}{dt}]_{others} & (m = 1) \\ [\frac{dc}{dt}]_{others_m} - RMSE_m \cdot \left[\frac{dRMSE}{d[\frac{dc}{dt}]_{others}}\right]_m^{-1} & (m > 1) \end{cases} \tag{30}$$

$$[\frac{dRMSE}{d[\frac{dc}{dt}]_{others}}]_m = [\frac{\Delta RMSE}{\Delta[\frac{dc}{dt}]_{others}}]_m = \frac{RMSE_m - RMSE_{m-1}}{[\frac{dc}{dt}]_{others_m} - [\frac{dc}{dt}]_{others_{m-1}}} \tag{31}$$

Where $m$ is the number of iterations and $\Delta[\frac{dc}{dt}]_{others}$ at the first iteration ($m = 1$) are estimated by Eq. (28). When the number

of iterations is greater than 1 ($m > 1$), the update equation of $\Delta[\frac{dc}{dt}]_{others}$ is developed base on New-Raphson method. The

*RMSE* is used as the objective function for optimization, and the derivative of *RMSE* with $[\frac{dc}{dt}]_{others}$ is estimated by the

Difference Method (DM) shown in Eq. (31).

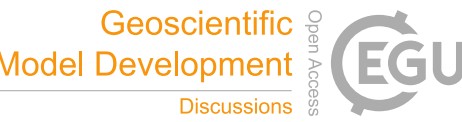

The $[\frac{dc}{dt}]_{others}$ can also be optimized in the mode of kinetic equations (*e.g.*, $[\frac{dc}{dt}]_{others} = k_{others} \times C$), and then the kinetic

constants ($k_{others}$) can be optimized using a similar process shown in Figure 1. ROMAC model provides an option for the user

to switch between these two modes. Furthermore, user can also use other algorithms for dynamic optimization, such as

Ensemble Kalman Filter (EnKF).

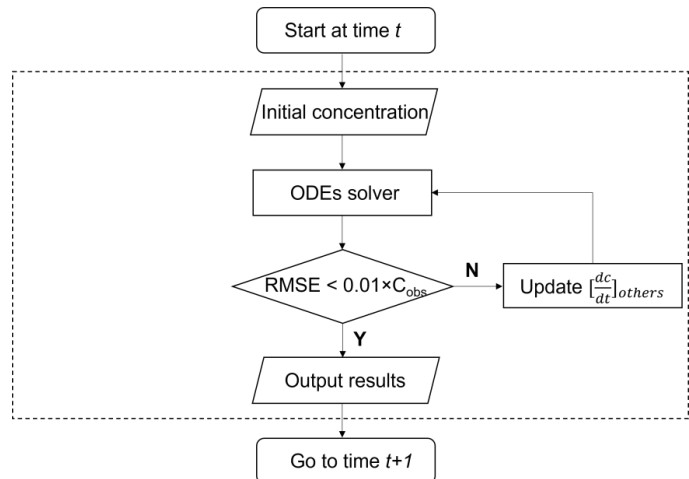

**Figure 1.** The cyclic dynamically optimization process of $[\frac{dc}{dt}]_{others}$.

**Scheme 2:**

In Scheme 2, the concentration of species can be initialized at the beginning of each simulation time step, which is mainly

applied to the solution of initial value problems and more suitable for chamber simulation. This scheme has been widely used

in previous models (*e.g.,* PBM-MCM, AtChem, F0AM). However, if the regional transport process of pollutants is not

considered, the simulation results of long-lived species in this scheme may have large deviations from the observed results.

**Scheme 3:**

Scheme 3 constrains the change rate of species concentration ($\frac{dc}{dt} = 0$) while constraining the initial concentration, in a similar

way as in F0AM. The advantage of this scheme is that the constrained variables can be kept at a user-specified level throughout

the simulation. In this scheme, the long-lived species can maintain the observed concentration level. This constraining is

appropriate if the temporal resolution of the observed data is high. The time interval of the model should be significantly

smaller than the lifetime of constrained species. However, this approach also has its limitations. Since species concentrations

are constrained as a constant, chemical imbalances may result in.

In order to better understand the characteristics of different constraint schemes, a simple running case was performed. Nitric

Oxide was constrained by three different schemes and exhibits its concentration variation characteristics. Figure 2 illustrates

the model output results of a test under different schemes to constrain species concentration. The output time step is 120



seconds, and the input interval of the observed data is 3600 seconds. In scheme 2, since emissions and regional transport are not considered, the concentration will decrease rapidly and then reach steady-state. Both scheme1 and scheme 3, the

concentration of NO in the model is well consistent with the observed hourly average concentration, which allows the state of the model to be consistent with the real atmospheric state. Due to different constraint schemes, the simulated concentrations of short-lived species (e.g., OH) will also differ. Users should choose a reasonable scheme according to the needs of their research and observation results.

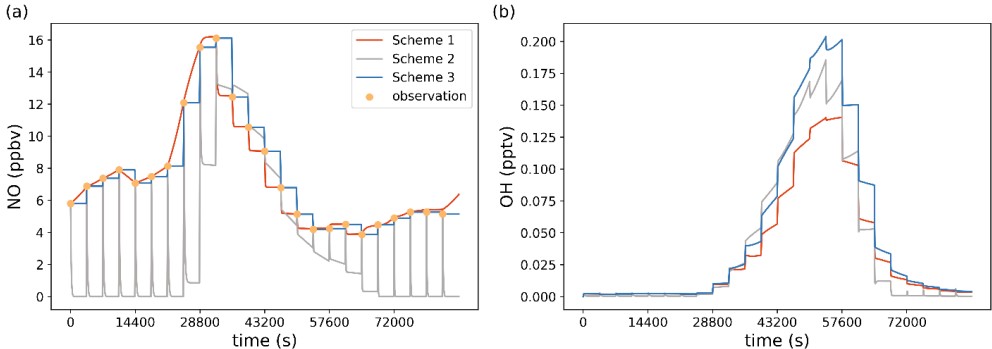

**Figure 2. Model output results under different concentration constraint schemes. (a) Variation characteristics of NO concentration in the model under different constraint schemes. (b) OH radical concentrations under different schemes corresponding to the left plot.**

**2.3 Photolysis**

ROMAC provides two ways for the user to set the photolysis rate. First, the user can specify the photolysis rate at each

integration time step in the form of an ASCII file. The input photolysis rate can be estimated by other models or the observations. In the ROMAC model, a python script (*TUV2ROMAC.py*) is provided for coupling the output of the Tropospheric Ultraviolet and Visible radiation model (TUVv5.2, available at https://www2.acom.ucar.edu/modeling/tropospheric-ultraviolet-and-visible-tuv-radiation-model). Users can easily use this tool to convert the TUV model output results into ROMAC input files.

ROMAC provides users with an inline calculation module to calculate photolysis. In the current version, the inline calculation module of photolysis uses the algorithm provided by MCM, an algorithm based on the solar zenith Angle (SZA). The trigonometric parameterization function is shown in Eq. (32). The parameters of *l, m, n* are provided by MCM (http://mcm.york.ac.uk/).

$$J = l \times \cos (SZA)^{m} \times e^{-n \times sec(SZA)} \tag{32}$$

If both the input photolysis rate and the inline calculated photolysis rate are present, ROMAC will use the input photolysis

rate preferentially. In addition, ROMAC provides the user with a photolysis rate modification factor (*Jrate*), users can easily



use this factor to adjust the photolysis rate in the model. The default value of *Jrate* is 1.0, and the actual photolysis rate used in the model is the input rate or the inline calculated rate multiplied by *Jrate*.

**2.4 Model accuracy and computational efficiency**

The comparison of ROMAC with AtChem, F0AM and FACSIMILE which is widely used for MCM was performed on a PC
with a CPU of 16-core AMD Ryzen 9 3950X at 3.5 GHz and 32 GB RAM. The computational efficiency of the model is evaluated by CPU time. AtChem, ROMAC, and PBM-MCM are all run using a single core and the CPU time is recorded by the software's built-in function. The CPU time used by F0AM is recorded by the function *cputime* in MATLAB. The total integration time is 259,200 seconds, and the integration time step is 900 seconds. The settings of *atol* ($10^{-4}$) and *rtol* ($10^{-3}$) in the models are consistent. The temperature, pressure and humidity in the scenario simulation are 25°C, 101.325 kPa and 35%,
respectively. The chemical mechanism used in this test is MCM v3.3.1, and the initial species concentrations are shown in Table A1. Since running the entire version of MCM v3.3.1 using AtChem is computationally excessive for our computing platform, we only selected the VOCs include in EPA's Photochemical Assessment Monitoring Stations (PAMS) Target List (https://www.epa.gov/amtic/ ) and exported the mechanism file from MCM website. In this test case, 3,899 species and 11,814 chemical reactions were included.

In this study, we assumed that the solution results of AtChem based on the CVODE library are accurate. Therefore, the accuracy of the model is evaluated by calculating the relative difference between the solution results of ROMAC and AtChem (Eq. (33)).

$$RE_t = \frac{|C_{ROMAC,t} - C_{AtChem,t}|}{|C_{AtChem,t}|} \times 100\% \tag{33}$$

Both EBI and VSVOR solvers in the ROMAC model are evaluated. Figure A1 shows a comparison of simulation results for nine species, including radicals and gaseous pollutants, which are commonly used in previous studies to evaluate solution
results (Hertel et al., 1993; Esentürk et al., 2018; Aro, 1996a). As shown in Figure A1, the solution results of ROMAC and AtChem are comparable, indicating that the solution results of ROMAC are comparable to the high-precision solution algorithm. Figure 3 illustrates the maximum relative error in the scenario simulation and the CPU time used by each model. The maximum relative errors between the results of the VSVOR solver and the results of AtChem are all smaller than the preset *rtol*. Compared with the single-step EBI solver, the VSVOR solver with variable time step and variable order can better
control the truncation error. Compared with the CPU time required to run, the VSVOR solver with higher solution accuracy is even more efficient than EBI. The CPU time consumed by EBI with different integration time steps is shown in Table A2. For the MCM chemical mechanism, the algorithm fails to converge when the integration time step is longer than 50 seconds. After



a series of tests, we found that even with an integration time step of 10 seconds, the EBI solver was at risk of failing to converge. Too small integration time step size makes EBI not efficient in solving MCM mechanism.

Compared with other models, ROMAC has greatly improved the computational efficiency of solving large-scale chemical mechanisms. The computational efficiency of ROMAC is 97% higher than that of F0AM and AtChem, and 96% higher than that of FACSIMILE.

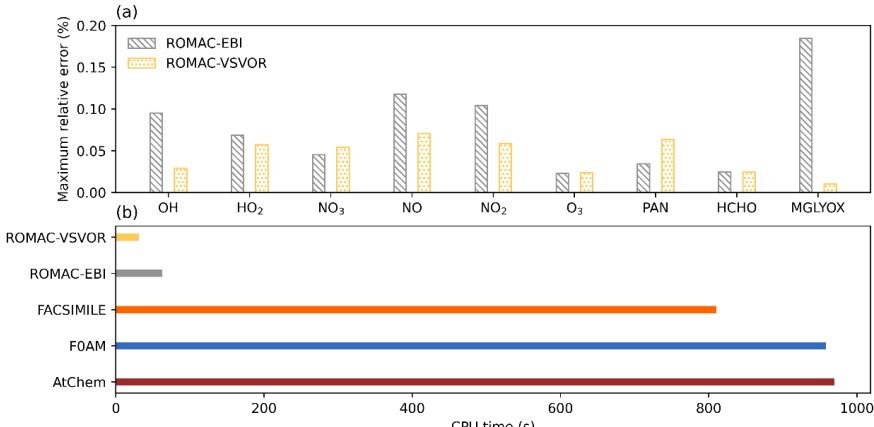

**Figure 3. Accuracy evaluation and comparison of model computational efficiency. (a) Maximum relative error between the**
**integration results of ROMAC and AtChem. (b) CPU time used to run compare with other models.**

### 3 Application of ROMAC model

#### 3.1 Chamber study

A chamber experiment for toluene degradation was used to evaluate the capabilities of ROMAC model to dynamically optimize chemical and physical processes. In this case, the indoor smog chamber in JNU-VMDSC was used to simulate the
degradation of toluene. The JNU-VMDSC provides a reliable experimental platform, and its structure and characterization (*e.g.,* wall loss, light intensity, airtightness test) have been described in previous study (Wang et al., 2023). Toluene and isoprene were injected into the chamber before the UV light was turned on. The initial mixing ratios of toluene and isoprene were 2,157 ppbv and 160 ppbv, respectively.

In order to simulate the effect of the dilution process on the toluene concentration, nitrogen was injected into the chamber
at a flow of 7 L/min, while sampling was carried out at a flow of 7 L/min at the sampling port. Similar to previous studies (Dada et al., 2020; Jiang et al., 2020), the rate of dilution was calculated using Eq. (34) and Eq. (35).

$$[\frac{dc}{dt}]_{dilu} = \frac{C \times dv}{V_{chamber} \times dt} = k_{dilu} \times C \qquad (34)$$





$$k_{dilu} = \frac{1}{V_{chamber}} \times \frac{dv}{dt} = \frac{Flow}{V_{chamber}} \qquad (35)$$

Where $k_{dilu}$ is the rate constant of dilution, $V_{chamber}$ is the volume of chamber (8000 L), and *Flow* is the flow of nitrogen injection. Therefore, the theoretically estimation result of $k_{dilu}$ in this case is $1.458 \times 10^{-5}$ s$^{-1}$. Wall loss was not considered in this simple experiment with gaseous pollutants.

The version of the chemical mechanism used in the model simulations is MCM v3.3.1, all species and mechanisms in MCM are included. Three scenarios case were set up to evaluate the simulation capabilities of ROMAC. In scenario 1, only chemical processes were considered (Eq. (36)). In scenario 2, chemical processes and dilution processes were considered (Eq. (37)). In scenario 3, we assume that the results of the experiment are influenced by an unknown process, and this process is assumed to be a first-order kinetic process (Eq. (38)). The $k_{others}$ in scenario 3 was dynamically optimized with scheme 1 as described in

Section 2.2. Theoretically, the value of $k_{others}$ obtained by the dynamically optimization in scenario 3 should be close to $k_{dilu}$ in scenario 2.

$$\frac{dc_{Tolu}}{dt} = \left[\frac{dc_{Tolu}}{dt}\right]_{chem} \qquad (36)$$

$$\frac{dc_{Tolu}}{dt} = \left[\frac{dc_{Tolu}}{dt}\right]_{chem} + \left[\frac{dc_{Tolu}}{dt}\right]_{dilu} = \left[\frac{dc_{Tolu}}{dt}\right]_{chem} + k_{dilu} \times C_{Tolu} \qquad (37)$$

$$\frac{dc_{Tolu}}{dt} = \left[\frac{dc_{Tolu}}{dt}\right]_{chem} + \left[\frac{dc_{Tolu}}{dt}\right]_{others} = \left[\frac{dc_{Tolu}}{dt}\right]_{chem} + k_{others} \times C_{Tolu} \qquad (38)$$

The total duration of the chamber experiment was 8 hours, and the CPU time consumed by a single simulation of ROMAC was about 13 seconds. Figure 4 illustrates the comparison results between the simulated and observed toluene mixing ratios for different scenario cases. Due to the lack of dilution process in scenario 1, there is a large gap between simulation results and observations. After considering the dilution process, the simulation of scenario 2 was improved, which indicates that the

setting of scenario 2 is reasonable. The simulation results of scenario 3 agree well with the observations, which indicates that the dynamic optimization algorithm successfully captures the process that cannot be explained by the MCM chemical mechanism. Figure 4b illustrates the chemical loss rate of toluene under different simulation scenarios. The results of scenario 3 and scenario 2 are consistent and significantly different from the results of scenario 1. This indicates that the dynamic

optimization algorithm can improve the chemical process while optimizing the physical process. Ignoring physical processes in the traditional box model may introduce large uncertainty to the simulation results. The optimized value of $k_{others}$ in scenario 3 is comparable to $k_{dilu}$ in scenario 2 (Figure 4c), which indicates that the dynamically optimized algorithm is reliable. Based on dynamic optimization, ROMAC can overcome the shortcomings of the over-simplified physical process in the traditional box model.




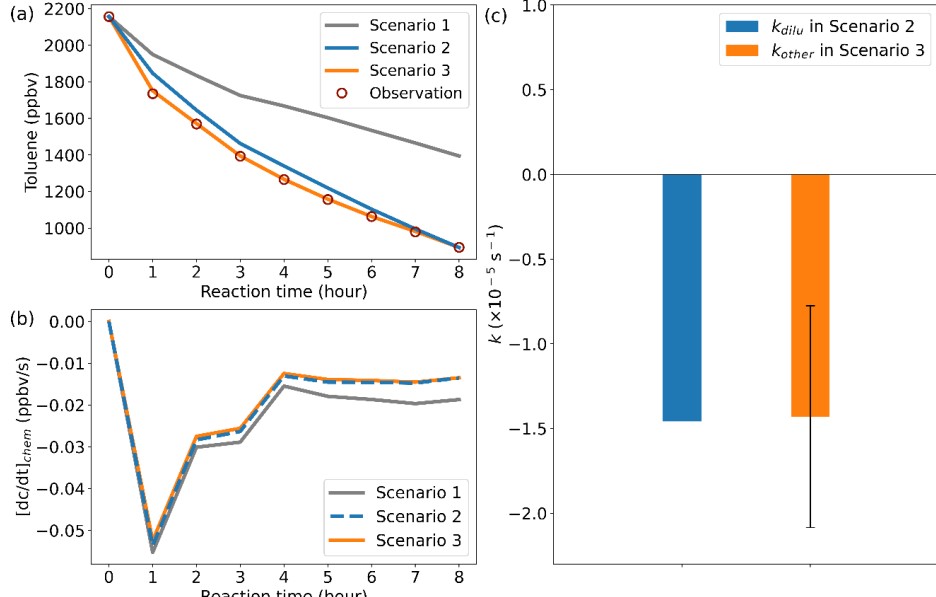

**Figure 4. Model simulation results. (a) Comparison results between the simulated and observed toluene mixing ratios. (b) Chemical loss rate of toluene. (c) Comparison of kinetic constants in dilution process. Error bars indicate the standard deviation of $k_{others}$ at different times in scenario 3.**


### 3.2 Field Observation

This case demonstrates the application of the ROMAC model to the analysis of the photochemical process of $O_3$ formation and the dynamical optimization of physical processes. The observation data were obtained at the Heshan Atmospheric Supersite (22.728°N, 112.929°E) in Guangdong Province, China. Detailed description of the Heshan site can be found in

previous publications (He et al., 2019; Yang et al., 2017). The observation period was from April 4, 2021 to April 10, 2021. Meteorological parameters and the mixing ratios of $NO_x$, VOCs, $SO_2$, CO were constrained by Scheme 3. The concentrations of $NO_x$ and VOCs were shown in Figure 5a, with the meteorological observations in Figure A2.

The simulation of $O_3$ was constrained by Scheme 1. In this case, all physical processes of $O_3$ (*e.g.,* dry deposition, dilution, transport) were merged into $[\frac{dc_{O_3}}{dt}]_{others}$. The rate of change of $O_3$ is shown in Eq. (39). The optimal estimate of $[\frac{dc_{O_3}}{dt}]_{others}$

uses the scheme 1 shown in Figure 1.

$$\frac{dc_{O_3}}{dt} = [\frac{dc_{O_3}}{dt}]_{chem} + [\frac{dc_{O_3}}{dt}]_{others} \qquad (39)$$





The comparison between the optimized simulation results and the observations of $O_3$ mixing ratios is shown in Figure 5b. As expected, the model outputs are consistent with the observations due to the dynamic optimization. The estimated value of $[\frac{dc_{O_3}}{dt}]_{others}$ for the physical process is shown in Figure 5c. Positive values of $[\frac{dc_{O_3}}{dt}]_{others}$ indicate that physical processes increase local $O_3$ concentration (*e.g.,* external transport), while negative values indicate that decrease $O_3$ concentration (*e.g.,*

dilution, deposition). As displayed in Figure 5c, $[\frac{dc_{O_3}}{dt}]_{others}$ is usually negative during the daytime, indicating that $O_3$ was transported out of the region after formation by photochemical processes. However, positive values of $[\frac{dc_{O_3}}{dt}]_{others}$ can also occur during the daytime. On April 6, the surface ozone mixing ratio increased rapidly, and the maximum hourly mixing ratio exceeded China II Emission Standard (>100 ppbv). The value of $[\frac{dc_{O_3}}{dt}]_{others}$ on the afternoon of April 6 is positive, indicating that physical processes were one of the reasons for the occurrence of $O_3$ pollution.

The rate of $O_3$ chemical production and precursor sensitivities were calculated using a method described in previous studies (Liu et al., 2022; Wang et al., 2018). As displayed in Figure 5d, the net $O_3$ production rate on April 6, 2021 was significantly higher than that on other days, indicating that chemical processes were also an important cause of $O_3$ pollution. The sensitivity of the $O_3$ formation to its precursors can be represented by relative incremental reactivity (RIR). Figure 6 shows the daily average RIR values of VOCs, $NO_x$ and CO. The RIR values of VOCs and CO were positive, which indicates that reducing the

concentration of VOCs and CO can effectively reduce the chemical formation of $O_3$. Except for April 8, the RIR values of $NO_x$ were negative, indicating that decreasing the $NO_x$ concentration leads to an increase in $O_3$ concentration. The negative values of RIR for NOx and higher positive values of RIR for VOC indicate that the ozone formation at the Heshan Atmospheric Supersite was mostly likely under VOC limited regime. The result was well consistent with a previous study (He et al., 2019), indicating that the application of ROMAC in chemical process diagnosis is reliable.

The application of this case demonstrates the ability of the ROMAC model to quantify the contribution of physical and chemical processes to air pollutant concentrations. Compared with the traditional Observation Based box Model (OBM), ROMAC overcomes the shortcomings of over-simplified physical modules. Compared with the emission-based 3D air quality model (*e.g.,* CMAQ, WRF-Chem, NAQPMS), the observation-based dynamic optimization algorithm in ROMAC model reduces the uncertainty introduced by emission inventory and meteorological simulation.



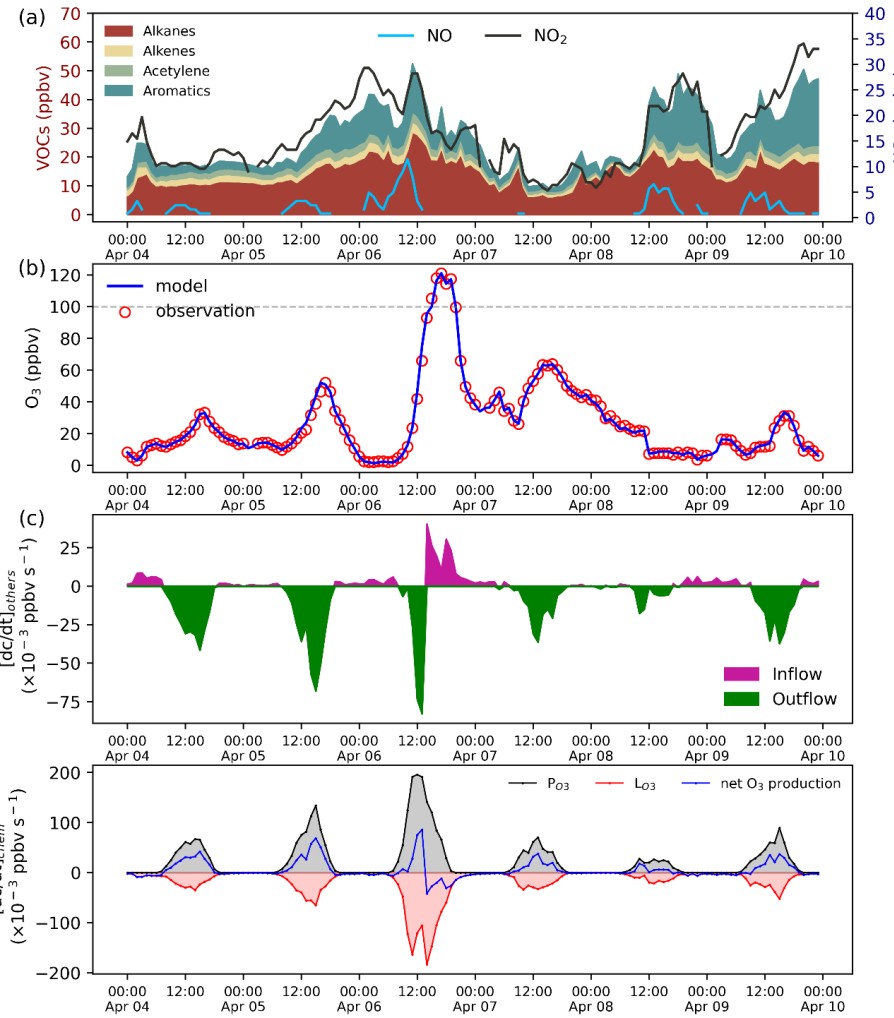


**Figure 5. Species mixing ratio and the rate of O₃ change. (a) VOCs and NOx mixing ratios. (b) Model and observation O₃ mixing ratios. (c) The effect of the physical process on the O₃ mixing ratios calculated by the adaptive dynamic optimization module. (d) The rate of O₃ chemical production.**

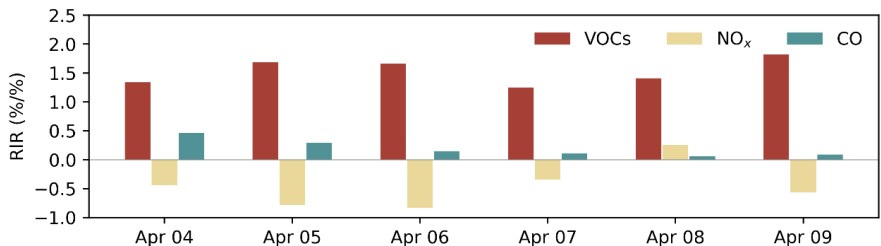


**Figure 6. RIR values of O₃ precursors, *i.e.*, VOCs, NOₓ and CO.**



## 4 Future development

The ROMAC model will be continuously updated and developed. Functionality for future improvements and upgrades,

includes

— A multiphase chemical reaction module and a module for Gas–particle partitioning and sectional simulation are being

developed.

— Adjoint sensitivity analysis will be added in a future version, and users can use ROMAC to analyze the relationship

between precursors and secondary pollutants.

— The Ensemble Kalman Filter (EnKF) will be added to dynamically optimize the physical process in future versions.





**Appendix A:**

**Table A1. Initial species concentration used for model comparisons (unit: molecules·cm⁻³)**

| Species | concentration | Species | concentration | Species | concentration |
|---------|---------------|---------|---------------|---------|---------------|
| O3 | 5.20E+10 | NC11H24 | 4.90E+08 | TM123B | 1.20E+09 |
| NO2 | 9.80E+11 | NC12H26 | 9.80E+08 | STYRENE | 5.90E+09 |
| NO | 9.80E+11 | C2H4 | 4.70E+10 | C4H6 | 4.90E+08 |
| CO | 1.50E+13 | C3H6 | 5.70E+09 | BENZAL | 1.90E+10 |
| SO2 | 7.50E+10 | BUT1ENE | 7.40E+08 | CH3COCH3 | 9.10E+09 |
| NO3 | 1.40E+08 | TBUT2ENE | 2.50E+08 | MEK | 4.90E+10 |
| C2H2 | 6.20E+10 | C5H8 | 2.50E+08 | | |
| C2H6 | 9.70E+10 | PENT1ENE | 1.80E+09 | | |
| C3H8 | 1.40E+11 | TPENT2ENE | 2.50E+08 | | |
| IC4H10 | 5.00E+10 | CPENT2ENE | 1.50E+09 | | |
| NC4H10 | 9.90E+10 | HEX1ENE | 2.50E+08 | | |
| IC5H12 | 1.30E+11 | TOLUENE | 1.00E+11 | | |
| NC5H12 | 1.70E+11 | BENZENE | 1.40E+10 | | |
| CHEX | 1.70E+09 | EBENZ | 4.20E+10 | | |
| M22C4 | 1.20E+09 | OXYL | 5.80E+10 | | |
| M23C4 | 7.20E+09 | IPBENZ | 1.50E+09 | | |
| M3PE | 7.40E+09 | PBENZ | 1.20E+09 | | |
| NC6H14 | 8.40E+09 | OETHTOL | 1.50E+09 | | |
| M2HEX | 6.60E+09 | METHTOL | 2.00E+09 | | |
| M3HEX | 6.40E+09 | TM135B | 2.50E+09 | | |
| NC7H16 | 4.40E+09 | HCHO | 1.20E+11 | | |
| NC8H18 | 4.40E+09 | CH3CHO | 3.90E+10 | | |
| NC9H20 | 4.40E+09 | C2H5CHO | 3.40E+09 | | |
| NC10H22 | 1.50E+09 | C3H7CHO | 1.70E+09 | | |
| PETHTOL | 2.00E+09 | MIBK | 3.80E+11 | | |
| TM124B | 2.20E+09 | HEX2ONE | 6.30E+11 | | |

**Table A2. CPU time used by the EBI solver at different integration time step sizes (unit: seconds). *Nonconvergence* represents that the EBI solver fails to converge.**

| Time step | 1 | 10 | 50 | 120 | 900 |
|-----------|---|----|----|-----|-----|
| CPU time | 182.8 | 59.7 | *Nonconvergence* | *Nonconvergence* | *Nonconvergence* |





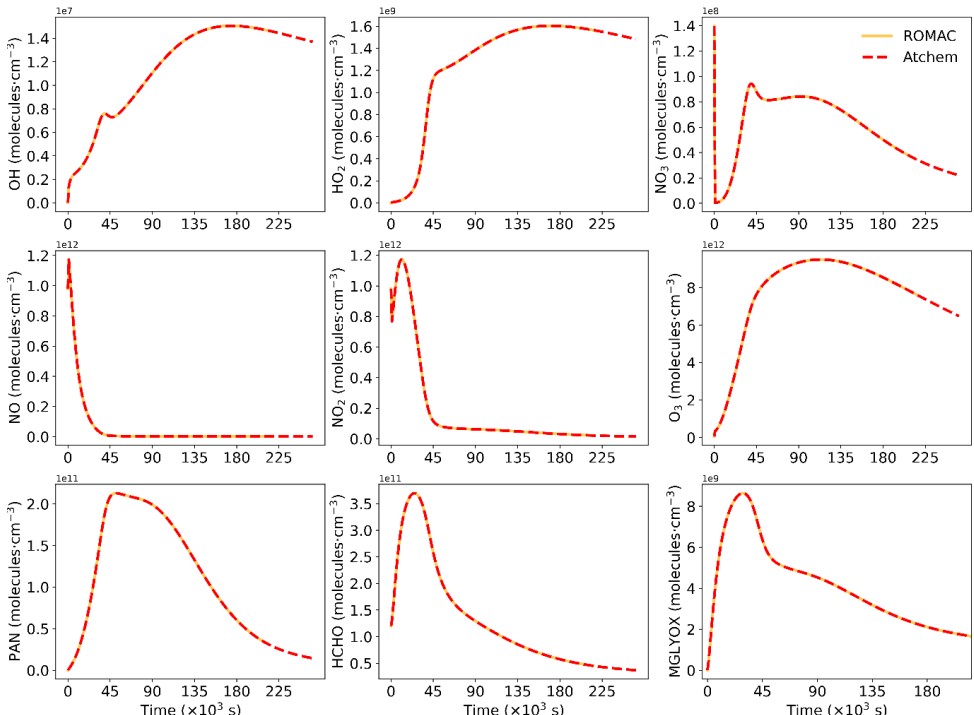

Figure A1. The integration results of the model comparison are compared with AtChem.




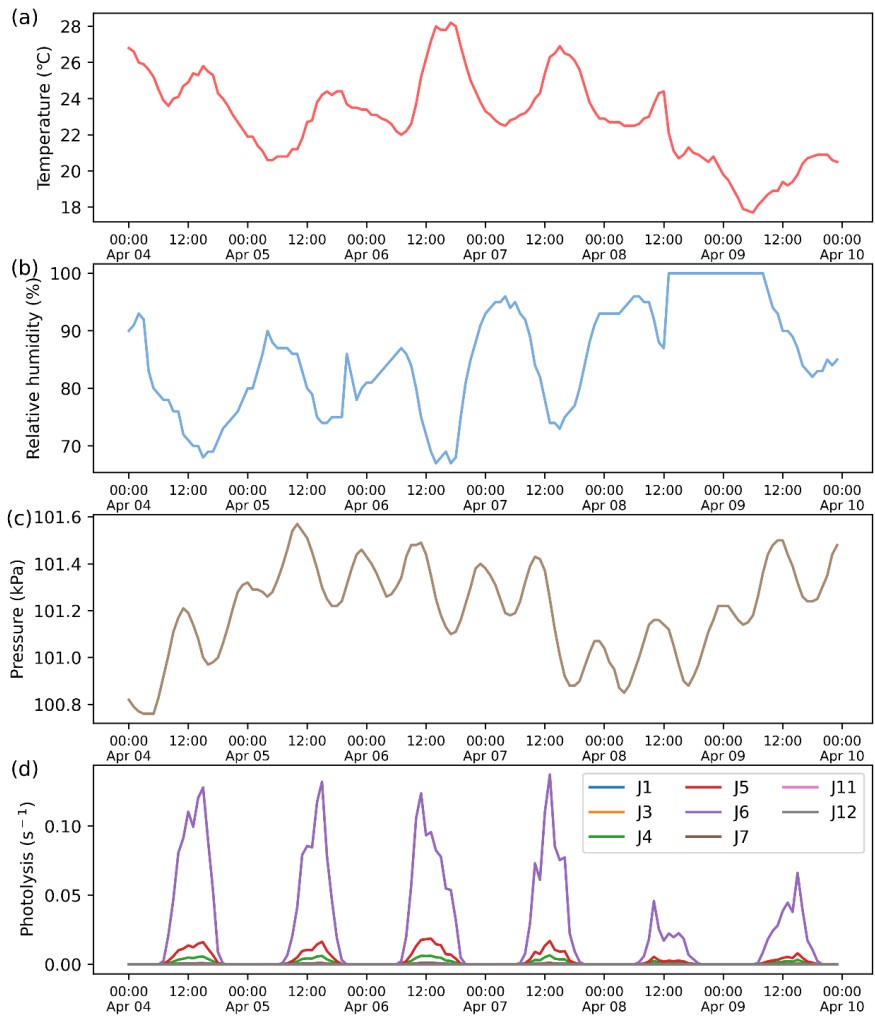

**Figure A2. Meteorological data input to the model. (a) temperature. (b) relative humidity. (c) atmospheric pressure. (d) photolysis rate**

*Code availability*

The current version of ROMAC coupled MCM v3.3.1 is archived on Zenodo: https://doi.org/10.5281/zenodo.7900781 under

the Attribution 4.0 International licence.

*Data availability*

The input data used to produce the results used in this paper is archived on Zenodo (https://doi.org/10.5281/zenodo.7900710).



*Author contributions*

**Jiangyong Li.** The developer of all model source code and algorithms for ROMAC; Conceptualization; Formal analysis;

Writing - Original Draft.

**Chunlin Zhang.** Formal analysis; Writing - Review & Editing.

**Wenlong Zhao.** Formal analysis; Software testing.

**Shijie Han.** The principal investigator of chamber study case; Data curation.

**Yu Wang**. Model Comparison and Evaluation.

**Hao Wang.** Funding acquisition; Writing – review & editing.

**Boguang Wang.** Funding acquisition; Writing – review & editing.

*Competing interests.*

The authors have declared no competing financial interest.


*Financial support.*

This work was supported by the National Natural Science Foundation of China (42121004, 42077190), and Science and
Technology Project of Guangdong Province of China (2019B121202002).

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
