# Peer review of "Rapid adaptive Optimization Model for Atmospheric Chemistry (ROMAC) v1.0"

_Geoscientific Model Development, 2023_

## Referee Comment (RC1)

**General comments**

In this paper, Li et al. present a new box model that demonstrates high computation efficiency with reasonable accuracy. The model's performance has been thoroughly evaluated through chamber experiments and in-situ observations, showcasing its capability to reproduce pollutant and radical concentrations under varying initial conditions. The results suggest the model could potentially benefit the modeling community. The paper is well-written and adequately referenced. There is only one general comment that I would like to pose. I commend the authors for discussing the implications of their box model in supporting more complex models, as mentioned in the Introduction. I am curious about the potential for easy adoption of the new model in existing CTMs or coupling with climate models. This aspect could significantly enhance the utility of the model if this is the case or it's planned in future developments.

**Specific comments**

1. Eq.1: It is essential to address why wet deposition is not included as a default item in the function, especially for hydrophilic components like sulfate. Providing an explanation or a discussion on this matter would add clarity to the model's capabilities.

2. Line145: The term "overcome" may not be suitable in describing the model's superiority over existing solvers, as it suggests that the issues present in other solvers have been completely resolved. Instead, consider rephrasing it to highlight that the new model offers an optimized algorithm that strikes a balance between efficiency and accuracy. Maybe also consider to replace the phrase throughout the paper.

3. Line184-185: 'the VSVOR solver has comparable computational efficiency with the EBI solver, and the solution accuracy and stability are better' – any obvious evidence on this than the equations listed above?

4. L232-233: It would be beneficial to offer a general recommendation on the choice of scheme for commonly studied species (e.g., $O_3$, PAN, $SO_2$) when utilizing the model. Users might find such a guide helpful when first implementing the model.

5. The model description section employs numerous abbreviations, which may hinder readability. I recommend creating a table containing all abbreviations to enhance the section's clarity and ease of understanding.

6. Figure A1: To improve clarity, consider using more distinct colors for the two models or converting one model to a scatter plot. Which solver is used to obtain the ROMAC results?

7. Line 285: The subtitle may not be suitable: it includes both model evaluation (esp. the chamber study section) than application.

8. Line 318-319: I'm not sure if such a conclusion can be drawn from Fig. 4c, as significant uncertainty exists in $k_{other}$.

---

## Author Comment (AC1)

**General comments**

In this paper, Li et al. present a new box model that demonstrates high computation efficiency with reasonable accuracy. The model's performance has been thoroughly evaluated through chamber experiments and in-situ observations, showcasing its capability to reproduce pollutant and radical concentrations under varying initial conditions. The results suggest the model could potentially benefit the modeling community. The paper is well-written and adequately referenced. There is only one general comment that I would like to pose. I commend the authors for discussing the implications of their box model in supporting more complex models, as mentioned in the Introduction. I am curious about the potential for easy adoption of the new model in existing CTMs or coupling with climate models. This aspect could significantly enhance the utility of the model if this is the case or it's planned in future developments.

**Response:**

Thank you very much for your comments. As of the current version (V1.0), ROMAC operates as a standalone model and does not offer integration capabilities with CTMs. However, in our future development roadmap, we have plans to introduce a modeling framework version of ROMAC known as "ROMAC-plug-in". The ROMAC-plug-in will provide the functionality to be called from Python or Fortran, while preserving its efficient design. This kernel will empower users to seamlessly construct their own models or integrate ROMAC into existing CTMs. We have included this development plan in the future development section in the revised manuscript.

Line412-416: In future development roadmap, we have plans to introduce a modeling framework version of ROMAC known as "ROMAC-plug-in". This ROMAC-plug-in will support calls from Python or Fortran, ensuring compatibility and flexibility for users. Importantly, the efficient design of ROMAC will be maintained, allowing for optimized performance. The kernel of ROMAC-plug-in will be specifically engineered to provide users with flexibility to effortlessly construct their own models or integrate ROMAC with existing frameworks, such as CTMs.

**Specific comments**

1. Eq.1: It is essential to address why wet deposition is not included as a default item in the function, especially for hydrophilic components like sulfate. Providing an explanation or a discussion on this matter would add clarity to the model's capabilities.

**Response:**

Thank you for your valuable suggestion. The primary focus of our box model simulation centers on chemical processes. In prior studies, the wet deposition process was often overlooked in the photochemical box model. Consequently, we did not develop a specific input interface for it. However, recognizing the significance of wet deposition in certain scenarios, we have included a freely definable rate term denoted as ($[\frac{dc}{dt}]_{others}$). This empowers users to introduce wet deposition into their simulations if deemed necessary. An explanation has been thoughtfully added to Section 2.

Line99-102: Note that the current version of ROMAC does not feature a dedicated input function for wet deposition. Instead, the ROMAC model allows users to set a custom rate term, $[\frac{dc}{dt}]_{others}$, which can be employed to account for wet deposition. If wet deposition is important for the simulation case, especially concerning the chemical mechanism of hydrophilic components like sulfate, it is suggested that the user incorporates it into $[\frac{dc}{dt}]_{others}$.

2. Line145: The term "overcome" may not be suitable in describing the model's superiority over existing solvers, as it suggests that the issues present in other solvers have been completely resolved. Instead, consider rephrasing it to highlight that the new model offers an optimized algorithm that strikes a balance between efficiency and accuracy. Maybe also consider to replace the phrase throughout the

paper.

**Response:**

Thank you for pointing out the improper application of words in the manuscript. We have used the term "outperform" to replace "overcome" in the revised manuscript. This sentence has been modified to emphasize the advantages of the VSVOR solver over the EBI solver:

Line171: Hence, this scheme will enhance the applicability and stability of the ROMAC numerical solver compared to the EBI numerical solver.

We modified the sentences that used the word 'overcome' throughout the paper.

Line14-16: ROMAC outperforms the traditional box models in evaluating the impact of physical processes on pollutant concentrations, and its ability to quantify the effects of chemical and physical processes on pollutant concentrations has been confirmed by the chamber and field observation cases.

Line84-85: Therefore, ROMAC will be computationally efficient and outperform the traditional box models in evaluating the impact of physical processes on pollutant concentrations.

Line391-392: Compared with the traditional Observation Based box Model (OBM), ROMAC is superior in evaluating the impact of physical processes on pollutant concentrations.

3. Line184-185: 'the VSVOR solver has comparable computational efficiency with the EBI solver, and the solution accuracy and stability are better' – any obvious evidence on this than the equations listed above?

**Response:**

Thanks for the comment. Yes, in addition to the equations, other evidences include a series of numerical experiments conducted to test the accuracy and stability of EBI and VSVOR which are presented in Section 2.4.

In terms of accuracy, in general, the VSVOR solver has second-order accuracy, and theoretically possesses a smaller truncation error compared to that of the EBI solver. In addition, we also evaluated the accuracy of the results by comparing with the results of AtChem (Figure 3). The differences between the VSVOR solver and AtChem results are all within the preset relative tolerance ($10^{-3}$). This is because that the VSVOR solver has a smaller truncation error and also has a strict error control scheme. However, the EBI solution results of some species (*e.g.,* NO, $NO_2$, MGLYOX) will exceed the preset relative tolerance.

In terms of stability, the fixed-step EBI solver may not converge due to the preset integration time step size being too large, which can be known from the test results in Table A2. The VSVOR solver with an adaptive variable time step size scheme can find the optimal integration time step and operating in a stable manner.

To reduce confusion, we have moved this conclusive statement to the end of Section 2.4 so that the readers can combine theoretical and numerical experimental results for a better understanding of the advances of the VSVOR solver.

Line 310-312: However, reducing the integration time step too much diminishes the efficiency of the EBI solver when handling the MCM mechanism in comparison to the VSVOR solver. Hence, the VSVOR solver exibits comparable computational efficiency to the EBI solver, while maintaining superior solution accuracy and stability.

4. L232-233: It would be beneficial to offer a general recommendation on the choice of scheme for commonly studied species (*e.g.,* $O_3$, PAN, $SO_2$) when utilizing the model. Users might find such a guide helpful when first implementing the model.

**Response:**

Thanks for your suggestion. We have incorporated the simulation results for PAN into Figure 2c. This inclusion provides reader with insight into how varying precursor constraint schemes can alter the simulation results for secondary pollutants. Users are

encouraged to select the most suitable scheme in accordance with their requirements. The scenarios in which each scheme can be applied have been described in Section 2.2. Additionally, we have rectified an error in the OH radical results that stemmed from a previous data processing mistake.

Line257-260: It is worth noting that due to variations in constraint schemes, simulated concentrations of other species, such as OH and PAN, can also diverge (Figure 2b and 2c). This case study was primarily designed to elucidate the unique features of different constraint schemes, with no intent to definitively validate or invalidate any particular scheme. Users are encouraged to make their scheme selections judiciously, aligning them with their specific research needs and observational findings.

[Figure]

**Figure 2. Model output results illustrating diurnal variations for selected species, highlighting the impact of different concentration constraint schemes. (a) NO concentrations; (b) OH concentrations; (c) PAN concentrations.**

5. The model description section employs numerous abbreviations, which may hinder readability. I recommend creating a table containing all abbreviations to enhance the section's clarity and ease of understanding.

**Response:**

Thank you for your valuable suggestion. To enhance the manuscript's readability, we have compiled a comprehensive list of abbreviations along with their corresponding descriptions in Table B1, which is included as an appendix.

Line107-108: The subsequent section offers a comprehensive overview of ROMAC's

features. Furthermore, to facilitate reference, all parameters employed in this paper are cataloged in Table B1.

**Table B1** Nomenclature

| Abbreviations | Explanation |
| --- | --- |
| ODEs | Ordinary Differential Equations |
| VSVOR | The variable-step and variable-order solver |
| $atol$ | absolute tolerance |
| $rtol$ | relative tolerance |
| r | The reactant in a chemical reaction |
| p | The product in a chemical reaction |
| α,β | Stoichiometric number |
| $C_t$ | Concentration of species at time $t$ |
| $f_i(C_{i,t}, t)$ | Rate of change of species $i$ at time $t$ |
| $P_{i,t}$ | Product rate of species $i$ at time $t$ |
| $L_{i,t}$ | Loss rate of species $i$ at time $t$ |
| $l_{i,t,R}$ | The part of the chemical reaction rate that is not directly related to the concentration of species $i$ in reaction $R$ at time $t$ |
| $\Delta t$ | Integration time step size |
| $g_1(C_{t+1})$ | The objective function when Newton's method solves the implicit Euler method |
| $g_2(C_{t+1})$ | The objective function when Newton's method solves the implicit trapezoidal method |
| $C_{t+1}^k$ | Species concentration at iteration $k$ of Newton's method |
| $\nabla g_1(C_{t+1})$ | The Jacobian matrix of $g_1(C_{t+1})$ |
| $\nabla g_2(C_{t+1})$ | The Jacobian matrix of $g_2(C_{t+1})$ |
| $\nabla g^{-1}(C_{t+1})$ | The inverse of the Jacobian matrix |
| $\Delta t_0$ | Integration time step size equal to $2.22 \times 10^{-16}$ s |
| $\Delta t_1$ | Minimum specie atmospheric lifetime in chemical mechanisms |
| $\Delta t_2$ | The maximum time step size necessary to achieve diagonal dominance of the Jacobian matrix. |
| $\Delta t_{init}$ | Initial integration time step size |
| $\Delta t_{max}$ | The maximum integration time step to ensure the result does not exceed the preset tolerance |
| $\Delta t_{opt}$ | Optimal integration step size |
| $RERR$ | Relative error calculated by the doubled-step method |
| $LTE$ | Local truncation error |
| $atol$ | Absolute tolerance |
| $rtol$ | Relative tolerance |
| $Rn$ | Lagrangian remainder in the Taylor expansion |
| ξ | Real number in the Lagrangian remainder in the Taylor expansion |
| $RMSE$ | Root Mean Square Error |

6. Figure A1: To improve clarity, consider using more distinct colors for the two models or converting one model to a scatter plot. Which solver is used to obtain the ROMAC results?

**Response:**

Thanks for your suggestion. We have now altered the representation of AtChem results to a dot shape. Also, the solver used by ROMAC in this test has been given the revised manuscript and also in the figure's caption.

Line298-299: The simulation results for ROMAC in Figure A1 are processed by the VSVOR solver.

[Figure]

**Figure A1. Comparison of the simulation results between ROMAC and AtChem for nine substances. ROMAC used the VSVOR solver in this test.**

7. Line 285: The subtitle may not be suitable: it includes both model evaluation (esp. the chamber study section) than application.

**Response:**

We appreciate your comment, and we have made the following adjustments accordingly:

- The subtitle of Section 3 has been modified to "Model Validation and Application."

- The subtitle of Section 3.1 has been updated to "Chamber Simulation Case."

- The subtitle of Section 3.2 has been revised to "Field Observation Case."

8. Line 318-319: I'm not sure if such a conclusion can be drawn from Fig. 4c, as significant uncertainty exists in $k_{other}$.

**Response:**

Thank you very much for pointing out this issue. The misleading statement, "Based on dynamic optimization, ROMAC can overcome the shortcomings of the over-simplified physical process in the traditional box model.", has been removed in the revised manuscript.

In this simulation case, the main reason that the observation cannot be reproduced in scenario 1 is that the physical process is missing in the model. This view can be proved by the fact that the simulation results better match the observation results after adding the theoretical calculation to the physical process in scenario 2. However, it is worth noting that there are still gaps between scenario 2 and the observations. Therefore, there should be uncertainty in the estimation of this physical process. But the expected value should be consistent with the theoretical calculated value. Dynamic algorithms incorporate fluctuations that cannot be captured by theoretical calculations into the results when calculating the effects of physical processes. Hence, the $k_{other}$ in scenario

3 have a range of fluctuations, however the average value is close to the theoretical calculation can prove that this scheme is feasible. A note on uncertainty has been added to the manuscript:

Line350-354: The rate of the physical process is subject to uncertainty in practical applications, but its average value is expected to closely approximate the theoretical value. The optimized value of $k_{others}$ in scenario 3, as shown in Figure 4c, exhibits a certain range of fluctuations rather than a fixed value. However, its average values ($1.430\times10^{-5}$) are comparable to $k_{dilu}$ in scenario 2 (Figure 4c), which indicates that the dynamically optimized algorithm is reliable.

---

## Author Comment (AC2)

Li et al. present a box model with a new photochemical solver, ROMAC v1.0, that aims to be flexible and computationally efficient through an unique adaptive dynamic optimization module. It also improves over traditional box models with better quantification of physical effects. A new chemical solver and box model development is exciting news for the field and well fit for the scope of GMD. I'm happy to recommend this manuscript for publication in GMD.

**Major comments:**

1. An important development for a highly efficient chemical solver is to move beyond the box model and improve the efficiency of 3-D chemical transport models. Could the authors elaborate on ROMAC's extensibility to be used, eventually, as a chemical solver component in 3-D models?

**Response:**

As described in Section 2.1, the VSVOR solver in ROMAC is a versatile chemical solver. Unlike the EBI solver in CMAQ, which is often customized for specific chemical mechanisms, the VSVOR solver is designed to be universal. This means that it possesses the capability to handle various chemical mechanisms in different models.

In the current version of ROMAC, the chemical solver is an integral part of the software and is not separately callable by the user. However, we are actively working on the development of the ROMAC-plug-in modeling tool. Please see our response to the general comment of the first anonymous referee for more details. This forthcoming tool will enable users to utilize the chemical solver program within ROMAC in their own models. Its design and development are currently underway, and we have added this development plan to the future development section in the revised manuscript.

Line412-416: In future development roadmap, we have plans to introduce a modeling framework version of ROMAC known as "ROMAC-plug-in". This ROMAC-plug-in will support calls from Python or Fortran, ensuring compatibility and flexibility for

users. Importantly, the efficient design of ROMAC will be maintained, allowing for optimized performance. The kernel of ROMAC-plug-in will be specifically engineered to provide users with flexibility to effortlessly construct their own models or integrate ROMAC with existing frameworks, such as CTMs.

In addition, the description and explanation of the extensibility of the chemical solver in ROMAC is also added.

Line111-114: This solver is engineered to enhance computational efficiency while accommodating the universal attributes of atmospheric chemical mechanism. It approaches all differential equations uniformly, eliminating the need for customized solution schemes tailored to specific chemical mechanisms. Therefore, the VSVOR solver offers a universal and versatile method for chemical solving.

Line152-154: The aforementioned characteristics are inherently present within the Jacobian matrix of chemical mechanisms and are impervious to variations in specific chemical mechanisms. As a result, this scheme proves to be universally applicable across different chemical mechanisms.

2. In the intro around L51, authors discuss simplified chemical mechanisms. These approaches have a long history. The authors mainly talk about "fixed" reductions of the chemical mechanism, where a larger mechanism (e.g., MCM) is processed down to fewer species beforehand and used. But there are general methods for reduction (e.g., Young & Boris, 1977; Djouad & Sportisse, 2002) and their on-line implementations (e.g., Sander et al., 2019; Shen et al. 2022; Lin et al. 2023) that provide stability and efficiency at some cost of error, and would be useful to include in the introduction.

**Response:**

Thanks for your comments, we have refined the parts of simplified chemical mechanisms and solution methods in the introduction section.

Line56-58: The simplified chemical mechanism can effectively improve the solution

efficiency of chemical processes, such as SAPRC07 (Carter, 2012), CB6 (Yarwood, 2010), MOZART(Emmons et al., 2010) and the Mainz Organic Mechanism (MOM) (Sander et al., 2019).

Line59-60: General methods for reduction (Young & Boris, 1977; Djouad & Sportisse, 2002) and their on-line implementations (Sander et al., 2019; Shen et al. 2022; Lin et al. 2023) had been developed.

3. More generally, I think that saying simplified mechanisms lead to bias to simulation results is not a fully fair statement to make. There is no one true chemical mechanism that gives 100% accurate answers - so is the bias defined against the MCM, or against the observations? Reduced mechanisms generally have a focus on getting particular parts of chemistry that are of interest correct, and thus introduce some degree of bias in species that are not fully represented (and it is unavoidable to have some bias in fast-cycling radicals).

**Response:**

Thanks for your comments, we wish to clarify that our intention is not to undermine the value of the simplification mechanism. Rather, we aim to convey that while the simplification mechanism can significantly enhance program efficiency, it cannot entirely substitute for the role of the near-explicit mechanism. Therefore, it remains imperative to enhance the efficiency of the near-explicit mechanism.

Furthermore, we recognize that the term '*bias*' may not be the most suitable choice. To address this concern, we wish to emphasize that our discussion revolves around distinguishing between a simplified mechanism and a near-explicit mechanism. According to your suggestion, we have revised the manuscript.

Line60-65: These simplified mechanisms are typically tailored to emphasize specific aspects of chemistry. As a result, the simulation results for certain species may diverge from those obtained using near-explicit chemical mechanism, particularly concerning

radicals (e.g., OH, $HO_2$, $RO_2$) and the concentrations of secondary pollutants (Ying and Li, 2011; Jimenez, 2003). The adoption of near-explicit chemical mechanisms enables a more detailed representation of the intricate process of photochemical reactions. Consequently, the simplified mechanism cannot adequately replace the role of the near-explicit mechanism.

4. The software is currently limited-access to reviewers. Will the software be open in the future / upon publication to GMD? That is very important to the community and I believe in line with GMD policy.

**Response:**

The current version of ROMAC coupled MCM v3.3.1 is archived on Zenodo: https://doi.org/10.5281/zenodo.7900781. This is mentioned in the code availability section. The software will be open access when the final manuscript is accepted and published in GMD. Future versions will continue to be published and open access on Zenodo, and updates will be noted in the current link.

**Minor comments:**

1. L71: The authors say that "Since the ROMAC model is computationally efficient, accurate and stable, users can dynamically optimize the influence of physical processes on pollutant concentration, and overcome the shortcomings of the lack of physical processes in the traditional box models." I'm not sure I follow here. What is the main difficulty in incorporating physical processes in other box models, is it a deliberate choice to focus on chemistry, or limited by efficiency, or (as the authors imply) affected by the stability of the solver? Incorporation of physical processes in a box model is a large part of the paper and I think the introduction would be better if more could be elaborated here, e.g., on why these processes weren't fully implemented and their effects.

**Response:**

We apologize for any confusion that may have arisen. One of the primary challenges is the limitations of 0-D box model, which lacks a three-dimensional structure. This limitation restricts the model's capability to directly simulate physical processes. This issue could be addressed by minimizing uncertainty in the chemical reactions while iteratively approximating the values associated with physical processes based on observations and simulations. However, this approach necessitates a model with high computational efficiency. We are confident that the ROMAC framework is well-suited for this task. In response to your suggestion, we have included these points into the introduction of the revised manuscript.

Line38-42: However, it is important to consider the impact of physical transport on long-lived species, such as its effect on $O_3$ concentration (Li et al., 2021; Liu et al., 2022). The 0-D model, which lacks a 3-D structure, is unable to directly estimate the impact of physical processes (e.g., vertical and horizontal transport) on pollutant concentrations. Therefore, it is necessary to find a proper scheme to estimate the physical process for these models.

2. L109-110: "...explicit methods ... are difficult to solve these problems." can be worded more clearly, "...explicit methods ... cannot achieve a stable solution without using a timestep shorter than all lifetimes in the system, which is computationally infeasible."

**Response:**

Thank you very much for your suggestions. We have revised the sentences in the manuscript.

Line130-132: Therefore, the ODEs system of atmospheric chemical kinetics simulation is extremely stiff, and explicit methods (*e.g.,* explicit Euler method, explicit Runge-

Kutta method) cannot achieve a stable solution without using a timestep shorter than all lifetimes in the system, which is computationally infeasible.

3. L130: The ROMAC model uses a Diagonal-Simplified-Newton (DSN) method which approximates the inverse of the Jacobian. Is there a quantified estimate of how much error this will introduce and effects on stability?

**Response:**

Similar to the Simplified-Newton method, the DSN method eventually converge to the result. The ROMAC model actively manages its computational precision. We add a supplementary note to this part. However, due to variations in initial conditions among different simulations, providing an exact quantification of its error is challenging. In Section 2.4, we assessed ROMAC's error control capabilities by comparing its results with those obtained using high-precision solvers.

Under the error control scheme, ROMAC can run stably. The occurrence of instability (*e.g.,* non-convergence or excessively large error) is typically attributed to a too large integration time step, and ROMAC will adaptively shorten the integration time step.

Line158-160: The solution process was iterated until the difference between the results of two iterations was less than one-tenth of the preset truncation error tolerance (*etc.,* $0.1 \times atol$ or $0.1 \times rtol$) for ODEs solution.

4. L255: Specify the OS version, compilers & versions used.

**Response:**

Thanks for your suggestion. The details information of OS version and compilers have been added to the manuscript.

Line282-283: The operating system was 64-bit Ubuntu (version 20.04.1) and the software was compiled using Intel Fortran (ifort version 2021.2.0).

5. In Section 2.4 authors show the accuracy of the model as expressed in maximum relative error %. Does the % grow over time throughout the integration? It would be useful to show a time series plot.

**Response:**

Thank you for your insightful suggestion. Ensuring stable error control is indeed crucial to the integrity of our work. To address this, we have updated Appendix A by including a time-series plot of the error in Figure A2. Additionally, the rate of change of the error over time is also added in Figure A2. As illustrated in Figure A2(a), the relative error gradually stabilizes rather than diverging over time. Further, Figure A2(b) demonstrates that after 225,000 seconds, the growth rate of the relative error is extremely minimal, falling within a range of $-1.0 \times 10^{-6}$ %/s to $1.0 \times 10^{-6}$ %/s.

Line300-303: The time series of relative error and its growth rates are depicted in Figure A2. The relative difference between the solution results of ROMAC and that of AtChem gradually stabilizes, and the rate of change of the relative error after 225,000 seconds is extremely minimal, falling within a range of $-1.0 \times 10^{-6}$ %/s to $1.0 \times 10^{-6}$ %/s.

[Figure]

**Figure A2. (a) Time series of relative errors, with dots marking the maximum values. (b) Growth rate of relative errors.**

6. In the abstract authors claim a 96% improvement in computational efficiency in ROMAC compared to "other box models". It may be useful to say which, and at what expense in error (which is small but worth mentioning).

**Response:**

Thanks for your suggestion, we made the following changes to the abstract to show that we did not sacrifice too much computational accuracy while improving efficiency.

Line16-20: Since the development of a variable-step and variable-order numerical solver that eliminates the need for Jacobian matrix processing, ROMAC's computational efficiency has seen a marked improvement with only a marginal increase in error. Specifically, ROMAC's computational efficiency has improved by 96% when compared to several established box models, such as F0AM and AtChem. Moreover, the solver maintains a discrepancy of less than 0.1% when its results are compared to those obtained from a high-precision solver in AtChem.

**Specific corrections:**

1. L132 "specie" -> "species"

**Response:**

Sorry for the syntax error, the following changes have been made.

Line156: According to the equations associated with the implicit Euler method in Eq. (1) to Eq. (13), the iteration formula for species $i$ is shown in Eq. (14).

**References:**

Carter, W. P. L.: SAPRC-07 CHEMICAL MECHANISMS, TEST SIMULATIONS, AND ENVIRONMENTAL CHAMBER SIMULATION FILES, 2012.

Djouad, R. and Sportisse, B.: Partitioning techniques and lumping computation for reducing chemical kinetics. APLA: An automatic partitioning and lumping algorithm, Applied Numerical Mathematics, 43, 383-398, https://doi.org/10.1016/S0168-9274(02)00111-3, 2002.

Emmons, L. K., Walters, S., Hess, P. G., and Lamarque, J.-F.: Description and evaluation of the Model for Ozone and Related chemical Tracers, version 4 (MOZART-4), Geosci. Model Dev.,, 3, 43-67, 2010.

Jimenez, P.: Comparison of photochemical mechanisms for air quality modeling, Atmospheric Environment, 37, 4179-4194, 10.1016/s1352-2310(03)00567-3, 2003.

Li, X. B., Fan, G., Lou, S., Yuan, B., Wang, X., and Shao, M.: Transport and boundary layer interaction contribution to extremely high surface ozone levels in eastern China, Environ Pollut, 268, 115804, 10.1016/j.envpol.2020.115804, 2021.

Lin, H., Long, M. S., Sander, R., Sandu, A., Yantosca, R. M., Estrada, L. A., Shen, L., and Jacob, D. J.: An Adaptive Auto‐Reduction Solver for Speeding Up Integration of Chemical Kinetics in Atmospheric Chemistry Models: Implementation and Evaluation in the Kinetic Pre‐Processor (KPP) Version 3.0.0, Journal of Advances in Modeling Earth Systems, 15, 10.1029/2022ms003293, 2023.

Liu, T., Hong, Y., Li, M., Xu, L., Chen, J., Bian, Y., Yang, C., Dan, Y., Zhang, Y., Xue, L., Zhao, M., Huang, Z., and Wang, H.: Atmospheric oxidation capacity and ozone pollution mechanism in a coastal city of southeastern China: analysis of a typical photochemical episode by an observation-based model, Atmospheric Chemistry and Physics, 22, 2173-2190, 10.5194/acp-22-2173-2022, 2022.

Sander, R., Baumgaertner, A., Cabrera-Perez, D., Frank, F., Gromov, S., Grooß, J.-U., Harder, H., Huijnen, V., Jöckel, P., Karydis, V. A., Niemeyer, K. E., Pozzer, A., Riede, H., Schultz, M. G., Taraborrelli, D., and Tauer, S.: The community atmospheric chemistry box model CAABA/MECCA-4.0, Geoscientific Model Development, 12, 1365-1385, 10.5194/gmd-12-1365-2019, 2019.

Shen, L., Jacob, D. J., Santillana, M., Bates, K., Zhuang, J., and Chen, W.: A machine-learning-guided adaptive algorithm to reduce the computational cost of integrating kinetics in global atmospheric chemistry models: application to GEOS-Chem versions 12.0.0 and 12.9.1, Geoscientific Model Development, 15, 1677-1687, 10.5194/gmd-15-1677-2022, 2022.

Wang, Y., Wang, H., Guo, H., Lyu, X., Cheng, H., Ling, Z., Louie, P. K. K., Simpson, I. J., Meinardi, S., and Blake, D. R.: Long-term O3–precursor relationships in Hong Kong: field observation and model simulation, Atmospheric Chemistry and Physics, 17, 10919-10935, 10.5194/acp-17-10919-2017, 2017.

Yarwood, G.: Development, Evaluation and Testing of Version 6 of the Carbon Bond Chemical Mechanism (CB6), 2010.

Ying, Q. and Li, J.: Implementation and initial application of the near-explicit Master Chemical Mechanism in the 3D Community Multiscale Air Quality (CMAQ) model, Atmospheric Environment, 45, 3244-3256, 10.1016/j.atmosenv.2011.03.043, 2011.

Young, T. R. and Boris, J. P.: A numerical technique for solving stiff ordinary differential equations associated with the chemical kinetics of reactive-flow problems, Journal of Physical Chemistry, 81, 2424-2427, https://doi.org/10.1021/j100540a018, 1977.

---

## Author Response (AR2)

The revised manuscript from Li et al. addresses all my prior comments and I am happy to recommend it for publication.

My only suggestion is that the experiment in Figure A2 for the relative errors appears to have peak errors for $O_3$ and $NO_3$ at the end of the simulation; I would suggest extending the experiment to see if there is further error growth or it can be constrained within a certain percentage. If errors continue to grow, I would suggest labeling the error growth rates in the bottom panel to ensure they are in a reasonable range. This is of particular interest to longer term atmospheric chemistry simulations.

**Response:**

Thanks for your suggestion. The numerical experiment was extended to a duration of 345,600 seconds. Before the end of the simulation, the error of $NO_3$ has peaked and remained stable. Although the relative error of $O_3$ has a trend of continue increase, the error growth rate was stable and extremely low ($3.3 \times 10^{-8}$ %/s). Hence, the relative error remains within the preset *rtol* even if the simulation duration is extended by an additional $2.0 \times 10^6$ seconds at this growth rate. We state the error growth rate for $O_3$ in the manuscript. The modifications to the manuscript are as follows.

**Line276-277:** The CPU time used by F0AM is recorded by the function *cputime* in MATLAB. The total integration time is 345,600 seconds, and the integration time step is 900 seconds.

**Line293-296:** Although the relative error of $O_3$ has a trend of continue increase, the growth rate of the error remains stable and extremely low ($3.3 \times 10^{-8}$ %/s). Hence, the relative error remains within the preset *rtol* even if the simulation duration is extended by an additional $2.0 \times 10^6$ seconds at this growth rate. This suggests that the ROMAC result's error can be stably controlled during long-term simulations.

[Figure]

**Figure 3. Accuracy evaluation and comparison of model computational efficiency. (a) Maximum relative error between the integration results of ROMAC and AtChem. (b) CPU time used to run compare with other models.**

**Table A2. CPU time used by the EBI solver at different integration time step sizes (unit: seconds).** *Nonconvergence* **represents that the EBI solver fails to converge.**

| Time step | 1 | 10 | 50 | 120 | 900 |
|---|---|---|---|---|---|
| CPU time | 256 | 73 | *Nonconvergence* | *Nonconvergence* | *Nonconvergence* |

[Figure]

**Figure A1. Comparison of the simulation results between ROMAC and AtChem for nine substances. ROMAC used the VSVOR solver in this test.**

[Figure]

**Figure A2. (a) Time series of relative errors, with dots marking the maximum values. (b) Growth rate of relative errors.**